# Evaluating Emotional Outcomes of Medical Students in Pediatric Emergency Medicine Telesimulation

**DOI:** 10.3390/children10010169

**Published:** 2023-01-15

**Authors:** Osamu Nomura, Momoka Sunohara, Ichiro Watanabe, Taichi Itoh

**Affiliations:** 1Department of Health Sciences Education, Hirosaki University, Hirosaki 036-8562, Japan; 2Centre for Community-Based Health Professions Education, Hirosaki University, Hirosaki 036-8562, Japan; 3Department of Emergency and Disaster Medicine, Hirosaki University, Hirosaki 036-8562, Japan; 4Department of Psychology, Concordia University, Montreal, QC H3G 1M8, Canada; 5Tokyo Metropolitan Children’s Medical Center, Division of Pediatric Critical Care Medicine, Tokyo 183-8561, Japan; 6Department of Emergency Medicine, University of Michigan, Ann Abor, MI 48109, USA; 7Department of Medical Education, University of Illinois at Chicago, Chicago, IL 60612, USA

**Keywords:** telesimulation, emotions, equivalent theory, COVID-19

## Abstract

The coronavirus disease 2019 (COVID-19) pandemic has challenged the feasibility of traditional in-person simulation-based clinical training due to the public health recommendation on social distancing. During the pandemic, telesimulation training was implemented to avoid multiple students and faculties gathering in confined spaces. While medical trainees’ perceived emotions have been acknowledged as a critical outcome of the in-person simulation-based training, the impact of telesimulation on trainees’ emotions has been unexamined. We conducted an educational team-based simulation study with a pediatric case of septic shock. Seventeen and twenty-four medical students participated in the telesimulation training and in-person simulation training, respectively. The institutional pandemic social restrictions at the time of each training session determined the participant assignment to either the telesimulation training or in-person simulation training. All participants responded to the Japanese version of the Medical Emotion Scale, which includes 20 items rated on a five-point Likert-type scale before, during, and after the simulation sessions. The measured emotions were categized into four emotion groups according to two dimensions: positive or negative and activating or deactivating emotions. The one-way analysis of variance between the telesimulation and in-person simulation training revealed no significant differences in the emotions perceived by the participants before, during, and after the simulation training sessions. The perceived emotions of medical students were comparable between the telesimulation and in-person simulation training. Further longitudinal studies with larger samples and multiple variables are needed to generalize the effectiveness of telesimulation.

## 1. Introduction

The 2019 coronavirus disease (COVID-19) pandemic has dramatically altered the landscape of clinical education for the next generation of healthcare professionals [1,2]. Restrictions and social distancing measures led many educational programs around the world to shift from in-person settings to online delivery modalities [3,4]. Likewise, it has been reported that the quality of medical education has also been negatively impacted by the changes and adjustments demanded by the COVID-19 measures. As social distancing was strongly recommended, medical educators have faced the challenge of not being able to provide medical students with sufficient in-person clinical training.

Traditionally, simulation education, as an educational method that encourages the acquisition of practical clinical skills in a safe environment outside of the workplace, has been shown to improve the quality of clinical education [5,6]. However, since the pandemic, remote and online learning has made it difficult to conduct conventional simulation education, based on in-person discussions and clinical skills practice, with many students and instructors gathered in a single space [7,8]. As a result, the implementation of telesimulation has enabled students and facilitators to learn from different places, and its effectiveness has been widely reported. Telesimulation is defined by McCoy [9] as “a process by which telecommunication and simulation resources are utilized to provide education, training, and/or assessment to learners at an off-site location” and is an innovative approach that typically involves learners and facilitators participating from different locations with the simulation equipment. As mentioned, the use of telesimulation boomed during the COVID-19 pandemic, which, in turn, brought methodological diversity to medical education. Typically, there are three types of telesimulation strategies: (1) students participate from their own home and are supervised by instructors from the simulation center; (2) students participate face-to-face as a simulation team at the simulation center and are supervised remotely by instructors; (3) all students and instructors access the simulation center from separate locations via a video conference system [7,8,10]. Each has its advantages and disadvantages, and educators choose an appropriate modality based on the strictness of the social distancing policy and the educational resources at each facility during the simulation activities. The current study applied the “students stay-at-home” strategy, as our participants were not fully vaccinated at the time of these educational activities.

The Simonson equivalence theory has received renewed attention in the wake of the COVID-19 pandemic [11,12]. The theory proposes that experiences in a distance education system are expected to be equivalent to face-to-face experiences. Therefore, students should achieve equivalent learning outcomes in both face-to-face and online settings. The equivalence theory supports our idea that students’ experiences, including emotions, would stay the same regardless of the instructional modalities (in-person or online) during the COVID-19 pandemic.

The effectiveness of simulation training has been examined through various variables based on educational frameworks such as the Kirkpatrick criteria [13]. The Kirkpatric model consists of levels one to four: (1) the trainees respond to their learning experiences; (2) learning outcomes, such as increased knowledge and skills; (3) changes in attitude towards their training; (4) improvements in the quality of their patient [14]. Evaluating the emotions of trainees is common in health profession education research utilizing the example of Kirkpatrick level 1 because the emotions are physiological and affective responses to the individuals’ experiences. In addition, emotions have attracted the attention of scholars in simulation-based medical education, as the development of emotion regulation skills in a critical situation is one of the desired competencies for trainees to acquire through the simulation session [15,16].

Furthermore, the study of achievement emotions [17], defined as “emotions tied directly to achievement activities or achievement outcomes”, focuses on their relevance to achievement activities and outcomes, which include diverse achievements in clinical work, such as solving clinical problems, performing surgical procedures, and communicating difficult news. Thus, the measurements based on the achievement emotions could be considered as the Kirkpatrick level 2. The most comprehensive and structured approach to interpreting achievement emotions is Pekrun’s control-value theory [18]. Pekrun’s control-value theory delineates the predictive relationships among distal and proximal antecedents, academic emotions, and student engagement and achievement. According to this theory, achievement emotions are classified into three dimensions: valence (positive versus negative), arousal (activating versus deactivating), and object-focus (activity-related versus outcome-related).

While learners’ perceived achievement emotions have been recognized as an important outcome in health professions, the impact of telesimulation on medical students’ emotions has not been examined.

### Objective

Therefore, the current study aimed to investigate the impact of telesimulation training on medical students’ emotions by comparing the perceived emotions of medical students in telesimulation with those of medical students in in-person simulation.

## 2. Materials and Methods

### 2.1. Design

This comparative simulation study used a team-based (six to seven members per group) pediatric scenario simulation. We compared the perceived emotions of Japanese medical students receiving telesimulation training with those of students receiving in-person training. The institutional COVID-19 pandemic social restrictions at the time of each training session determined the participant assignment to either a telesimulation training or an in-person simulation training. This environment therefore provided a quasi-randomization procedure in this study.

### 2.2. Procedures

In our “stay-at-home” telesimulation, students joined in the telesimulation from their home using Microsoft Teams. They were able to view a simulation room where the instructor and infant mannequin were presented. Before each session, the instructors presented a case scenario to the students. During the session, students asked the instructor to perform primary and secondary surveys on the infant mannequin, and the instructor informed the students of the survey findings. The vital signs of the scenario were shown on the screen of the simulation room using Microsoft Teams and the Sim monitor app, and the instructor changed the vital sign of the scenario based on the interventions the students asked the instructors to perform.

We set the simulation scenario as an infant case of septic shock because pediatric septic shock management, including the recognition of sepsis, fluid resuscitation, and early administration of antibiotics, is an essential skill in pediatrics and emergency medicine at our university [19]. This “students stay-at-home” telesimulation focused on promoting students’ clinical reasoning skills in pediatric emergency medicine rather than technical skills such as physical examinations, cardiopulmonary resuscitation, and airway management. In addition, we prioritized developing students’ cognitive skills (i.e., clinical reasoning) because providing students staying at home with engagement in the cognitive processes needed by health professionals facilitates their professional identity formation even during the COVID-19 pandemic [20,21].

### 2.3. Participants

Fifth-year medical students at Hirosaki University in Japan who had a clinical rotation training at the emergency department were eligible to participate in this study. We conducted our study during October 2020–March 2021. Of 41 students, 17 students (three teams) and 24 students (four teams) were assigned to the telesimulations and the in-person simulations, respectively. This group assignment was determined based on the government and university’s lockdown policy during the COVID-19 pandemic. All participants completed the Japanese version of the Medical Emotion Scale (J-MES)—a Japanese translation of the Medical Emotion Scale originally developed in English (Figure 1) [22]. The Medical Emotion Scale was developed based on the control-value theory of achievement emotions [17,18].

### 2.4. Data Collection

The J-MES consists of 20 items containing adjectives describing discrete emotions, and uses a five-point Likert scale. The items are categorized into four subscales (Figure 2) according to the valence (positive/negative) and arousal level (activating/deactivating) of the emotions: (a) positively activating (e.g., happiness, hope); (b) positively deactivating (e.g., relaxed, relieved); (c) negatively activating (e.g., anger, shame), or (d) negatively deactivating (e.g., sadness, boredom). In addition, the “during” questionnaire was administered immediately after the task to prevent interference with the task.

### 2.5. Analysis

A series of two-way mixed ANOVA was conducted on the influences of two independent variables (emotion type and experimental condition) on the specific levels of assessed emotions before, during, and after the task. Emotion type included four levels: positive activating emotions (PA), positive deactivating emotions (PD), negative activating emotions (NA), and negative deactivating emotions (ND). Our experimental condition factor consisted of two levels: the simulation group and the traditional group. We used the 1000 resampling bootstrapping method to give us a better estimate.

Cronbach’s alpha coefficients of the internal consistencies of the J-MES measured before, during, and after the task were 0.82, 0.75, and 0.76, respectively, all of which reached satisfactory levels in the same range as the criteria recommended in the existing literature. All the data were analyzed using JASP, version 0.16.

### 2.6. Ethics

The Graduate School of Medicine Ethics Committee, Hirosaki University, approved this study. Written informed consent was obtained from the participants before they participated in the study.

## 3. Results

Table 1 shows the descriptive statistics of the participants.

### 3.1. Emotions before the Task

The results (Table 2) showed that there was a significant main effect of emotion type (*F* (2, 62) = 28.70, *p* < 0.001, η*_p_*^2^ = 0.424), with participants reporting different levels of emotions for the four emotion types: PA (*M* = 3.06), PD (*M* = 2.63), NA (*M* = 1.72), and ND (*M* = 1.95). There was no significant main effect of experimental condition (*F* (1, 39) = 0.215, *p* = 0.65, η*_p_*^2^ = 0.01) on assessed emotions. Our results also showed that there was no significant interaction between emotion type and experimental condition (*F* (2, 62) = 0.15, *p* = 0.613, η*_p_*^2^ = 0.01)

### 3.2. Emotions during the Task

The results (Table 3) showed that there was a significant main effect of emotion type (*F* (2, 65) = 41.17, *p* < 0.001, η*_p_*^2^ = 0.51), with participants reporting different levels of emotions for the four emotion types: PA (*M* = 3.33), PD (*M* = 2.94), NA (*M* = 1.85), and ND (*M* = 1.51). There was no significant main effect of experimental condition (*F* (1, 39) = 0.34, *p* = 0.56, η*_p_*^2^ = 0.01) on assessed emotions. Our results also showed that there was no significant interaction between emotion type and experimental condition (*F* (2, 65) = 0.37, *p* = 0.654, η*_p_*^2^ = 0.01).

### 3.3. Emotions after the Task

The results (Table 4) showed that there was a significant main effect of emotion type (*F* (2, 75) = 51.11, *p* < 0.001, η*_p_*^2^ = 0.57), with participants reporting different levels of emotions for the four emotion types: PA (*M* = 3.39), PD (*M* = 3.23), NA (*M* = 1.66), and ND (*M* = 1.50). There was no significant main effect of experimental condition (*F* (1, 39) = 1.24, *p* = 0.27, η_*p*_^2^ = 0.03) on assessed emotions. Our results also showed that there was no significant interaction between emotion type and experimental condition (*F* (2, 75) = 0.45, *p* = 0.631, η_*p*_^2^ = 0.01).

## 4. Discussion

To our knowledge, this is the first study examining the impact of telesimulation on medical students’ perceived emotions. We demonstrated that medical students’ perceived emotions did not differ between telesimulation and in-person simulation, which was supported by the Simonson equivalence theory. These findings add to evidence of the effectiveness of telesimulation from medical students’ perspectives on affective outcomes (i.e., emotions).

We measured the students’ emotions as the primary outcome of the study. Cognitive science suggests that learners’ emotions modulate perception, memory, attention, and performance, including cognitive reasoning and psychomotor skills, suggesting that learners’ emotions are an essential predictor of performance [23]. In addition, the concepts of learner-centeredness in educational psychology currently propose that emotions can be defined as a “surrogate outcome” in research because performance in simulated settings does not necessarily reflect real-world performance [24]. Recent simulation studies, therefore, have adopted emotion measurements as study outcomes for evaluating the effectiveness of educational strategies [15,16]. The finding from our study was consistent with the equivalence theory [11,12], with the students experiencing equivalent emotions across the two teaching modalities.

Telesimulation has been used to provide simulation training in resource-limited settings; however, the use of telesimulation has expanded during the COVID-19 pandemic to retain the learning opportunities for medical trainees under the social distancing regulations. For example, Naik et al. reported that interactive telesimulation combined with a video tutorial effectively provided practical knowledge on ventilator management for COVID-19 patients for the non-ICU healthcare providers with limited experience who were urgently redeployed to treat COVID-19 patients in ICU [25]. In surgery, telesimulation for teaching the residents advanced laparoscopic suturing improved the trainees’ performance not only in simulation settings but also in the operation room [26]. A study conducted in undergraduate critical care medicine showed that virtual telesimulation interprofessional sepsis team training effectively facilitated the acquisition of sepsis knowledge and communication skills in medical and nursing students [27]. The current study newly showed the educational effectiveness of telesimulation on medical emotional outcomes in the field of pediatric emergency medicine.

### Limitations

Several limitations are noted in this study. First, there is a risk of type II error due to the small sample size in our study. However, we could not continue collecting samples for the telesimulation group after the pandemic situation improved from the educational perspective. Second, only a single scenario was used for the simulation; thus, the generalizability of our finding is uncertain. Third, we were not able to control the communications among the students of the two groups during the study period. Thus, there is a risk of information bias due to information transmission from one group to another; however, it was not justified that we prevent the students from communicating with each other for 6 months for this study purpose. Forth, we did not calculate the desired sample size for this study as there has not been a previous study on telesimulation using the emotion scale and the effect size was uncertain. Finally, we evaluated only one subjective variable (i.e., emotions) based on the self-reported questionnaire; thus, a reporting bias could have affected the results. Therefore, other objective measurements, such as clinical reasoning performance tests, are needed in pediatrics [28]. In addition, the data on emotions were collected at one point of the study, and long-term psychological outcomes such as wellness remain uncertain [29]. Further longitudinal studies with larger samples and multiple variables are needed to address these limitations.

## 5. Conclusions

This study demonstrated that medical students perceived equivalent emotions between telesimulation and in-person simulation during the COVID-19 pandemic. Other evidence from different types of telesimulation needs to be examined.

## Figures and Tables

**Figure 1 children-10-00169-f001:**
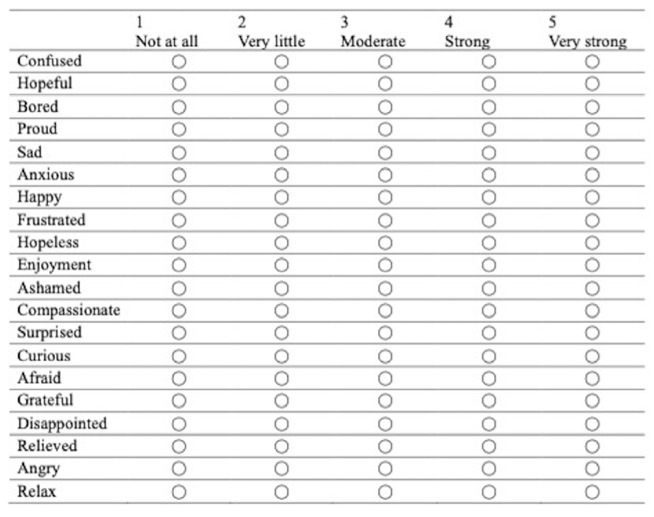
Sample of the Medical Emotion Scale.

**Figure 2 children-10-00169-f002:**
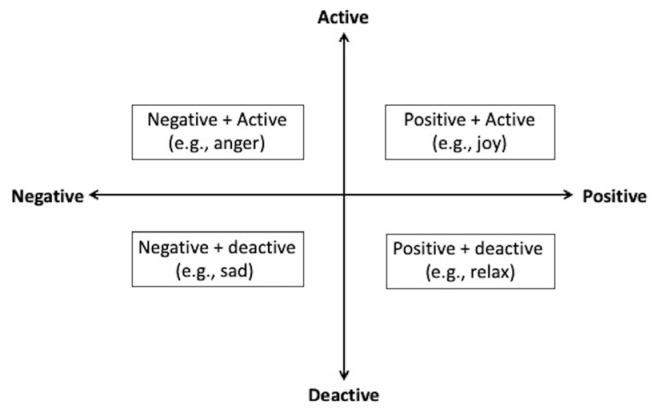
Category of emotion items.

**Table 1 children-10-00169-t001:** The descriptive statistics (i.e., before, during, and after the task).

	Telesimulation (*n* = 17)	In-Person Simulation (*n* = 24)
Before	M (SD)	M (SD)
Pos. act. emotions	2.98 (0.80)	3.14 (0.74)
Pos. deact. emotions	2.56 (0.59)	2.71 (1.00)
Neg. act. motions	1.89 (0.68)	1.99 (0.67)
Neg. deact. emotions	1.79 (0.79)	1.64 (0.67)
During	M (SD)	M (SD)
Pos. act. emotions	3.18 (0.89)	3.43 (0.88)
Pos. deact. emotions	2.82 (1.07)	3.02 (1.1)
Neg. act. emotions	1.86 (0.75)	1.85 (0.64)
Neg. deact. emotions	1.56 (0.58)	1.47 (0.68)
After	M (SD)	M (SD)
Pos. act. emotions	3.33 (0.92)	3.43 (0.95)
Pos. deact. emotions	3.32 (0.86)	3.17 (1.04)
Neg. act. emotions	1.77 (0.69)	1.58 (0.73)
Neg. deact. emotions	1.69 (0.63)	1.36 (0.80)

Note. M, mean; SD, standard deviation.

**Table 2 children-10-00169-t002:** ANOVA results of emotions before the task.

Variables	Sum of Squares	*df*	Mean Square	*F*	*p*	Partial η^2^
(Intercept)	870.70	1	870.70	1189.11	0.000	0.97
Telesimulation	0.16	1	0.16	0.22	0.646	0.01
Emotions	46.09	3	15.36	28.70	0.000	0.44
Telesimulation × Emotions	0.67	3	0.22	0.42	0.741	0.01
Error	62.63	39	0.54			

**Table 3 children-10-00169-t003:** ANOVA results of emotions during the task.

Variables	Sum of Squares	*df*	Mean Square	*F*	*p*	Partial η*^2^*
(Intercept)	916.66	1	916.66	1112.46	0.000	0.966
Telesimulation	0.28	1	0.28	0.34	0.563	0.009
Emotions	86.67	3	28.89	41.17	0.000	0.514
Telesimulation × Emotions	0.78	3	0.26	0.37	0.774	0.009
Error	82.09	39	0.70			

**Table 4 children-10-00169-t004:** ANOVA results of emotions after the task.

Variables	Sum of Squares	*df*	Mean Square	*F*	*p*	Partial η*^2^*
(Intercept)	960.85	1	960.85	1496.79	0.000	0.975
Telesimulation	0.79	1	0.79	1.24	0.273	0.031
Emotions	116.83	3	38.94	51.11	0.000	0.567
Telesimulation x Emotions	1.03	3	0.34	0.45	0.717	0.011
Error	89.15	39	0.76			

## Data Availability

The data supporting this study’s findings are available from the corresponding author upon reasonable request.

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
