# Peer review of "Evaluating Emotional Outcomes of Medical Students in Pediatric Emergency Medicine Telesimulation"

_children, 2023, doi:10.3390/children10010169_

Round 1

Reviewer 1 Report

Dear Authors,

I found your work quite interesting in the topic, but very poorly developed.

From the introduction, I found the there is no sufficient background, and it should be extended by talking more broadly about the premises, such as the Kirkpatrick criteria or the Simonson equivalence.

 Secondly, it is not clear what the true meaning of emotion valence and activation is, or it is very difficult to understand from the paper, especially for people who are not into these topics. Thus, it is unclear how the emotions were related to the scenario: for example, what was the initial scenario? What was the specific task the students faced? And so on.

 Most importantly, the sample size is far too small (17 and 24 students), so the statistical correlations are very weak.

Moreover, the discussion section is not well organized, as it includes a paragraph, the second one, which should be inserted in the introduction section. Furthermore, the correlation and discussion related to other simulation studies is too brief and needs to be expanded, by discussing in more detail the previous studies and how they relate to the present study.

Finally, I found the English presentation to be improved, as it is not very clear in some passages.

Author Response

Comment 1

From the introduction, I found the there is no sufficient background, and it should be extended by talking more broadly about the premises, such as the Kirkpatrick criteria or the Simonson equivalence.”

Response

Thank you for your important pointing out. As you suggested, we extended the descriptions of the Kirkpatrick criteria, the Simonson equivalence, achievement emotion, and the control value theory in the background. (Line 71-103)

Comment 2

Secondly, it is not clear what the true meaning of emotion valence and activation is, or it is very difficult to understand from the paper, especially for people who are not into these topics. Thus, it is unclear how the emotions were related to the scenario: for example, what was the initial scenario? What was the specific task the students faced? And so on..

Response

Thank you for your suggestions. We first change the term activation to arousal level to clarify the definitions and display the figures (Figure 1 & 2) of the J-MES to promote the readers’ understanding (L 154). Also, we added descriptions of why emotions are important in the simulation training of undergraduate medical students in the introduction (L83-100). Measuring emotions are not specific to this scenario of septic shock but just examining students’ emotions as the study measurements of interests. Finally, we measured medical students' emotions while the student team performed the resuscitation for a septic infant case in simulation.

Comment 3

Most importantly, the sample size is far too small (17 and 24 students), so the statistical correlations are very weak.”

Response

Thank you for your critical comments. The small sample size is a significant limitation of this study.

However, telesimulation is a new field of pediatric medical education, and our study could add new evidence to the body of knowledge in this field, even if the statistical effect size is small. We have written this issue in the limitation section.

Comment 4

Moreover, the discussion section is not well organized, as it includes a paragraph, the second one, which should be inserted in the introduction section.

Response

Thank you for your insightful comment. As you suggested, we moved the second paragraph of the discussion section to the introduction (L61-70).

Comment 5

Furthermore, the correlation and discussion related to other simulation studies is too brief and needs to be expanded, by discussing in more detail the previous studies and how they relate to the present study...

Response

Thank you for your pointing this out. We added detail descriptions of the previous studies on other simulation studies (L223-236).

Comment 6

Finally, I found the English presentation to be improved, as it is not very clear in some passages.

Response

Thank you for your suggestion. English editing was performed.

Reviewer 2 Report

Firstly, thank you for opportunity to review very interested article. I don't feel qualified to judge about the English language and style due to not native language.

1. The title reflect the main subject about outcome of medical student in telesimulation, title was clear and easy to understand.

2. The abstract summarize and reflect the work described in the manuscript.

3. The key words reflect the focus of the manuscript.

4. The manuscript adequately describe the background, present status, and significance of the study. The authors explain adaptation of medical student learning process during COVID-19 pandemic.

5. The manuscript describe methods in adequate detail, study subjects were clear, with demonstrate IRB number or text to human ethics consideration. I suggest the authors explain about "contamination" between two groups. The participants was divided into 2 groups and possible contamination between that. Some of members in telesimulation can contact in other group and talk about learning process, in this case, how method to avoid contamination?

- inclusion and exclusion criteria ?

- sample size calculation ?

- How method to divides participants into each group ?

6. The research objectives achieved by the experiments used in this study. 

7. The manuscript interpret the findings adequately and appropriately, highlighting the key points concisely, clearly, and logically.

8. Tables and figures sufficient, good quality and appropriately illustrative of the paper contents.

9. The manuscript meet the requirements of biostatistics.

10. The manuscript cite appropriately the latest, important, and authoritative references in the introduction and discussion sections. 

Author Response

Comment 1

I suggest the authors explain about "contamination" between two groups. The participants was divided into 2 groups and possible contamination between that. Some of members in telesimulation can contact in other group and talk about learning process, in this case, how method to avoid contamination?

Response

Thank you for your critical comments. We did not prevent the interactions between the students of tele-simulation and in-person simulation groups; thus, there was a potential for contamination. The simulation session occurred every two weeks for a total of 6 months, and it was almost impossible to avoid contact of the students of the two groups for a long period. We, therefore, write this issue in the limitation section (L242-248).

Comment 2

- inclusion and exclusion criteria ?

Response

The inclusion criteria are the 5th-year medical student who participated in the pediatric emergency medicine simulation training during the study period. The exclusion criteria are the students who disagreed with participating in this study. However, all of the eligible students consented to participate in this study. We added this description in the method (L141-146).

Comment 3

- sample size calculation ?

Response

We did not do sample size calculation for this study as we considered this study as an exploratory trial. However, since there are very few studies on telesimulation in pediatric emergency medicine using an emotion scale, it was not possible to estimate the effect size the impact of telesimulation on students' emotions (L246-248).

Comment 4

- How method to divides participants into each group ?”

Response

The medical students who rotated to the clinical clerkship of emergency medicine during the lockdown policy of the university were allocated to the tele-simulation groups. The other students were allocated to the in-person simulation group (L145-146).

Round 2

Reviewer 1 Report

Dear authors

Thanks for accepting the suggestions and editing the parts of the text that were unclear. It is also evident that the text has been reviewed by an english native speaker.

Now the text is much more understandable and the various sections are very well developed.

 Although the theme is innovative, the study well designed and well displayed, I still find that there are great limitations to your work. Among the limits, as already mentioned, there is the small sample size of students enrolled in the study, which make the statistical data very weak. Moreover, the study was conducted on a single scenario, while it would have been useful to analyze various clinical settings to increase the data obtained about students' emotions during telesimulations.

Unfortunately these data cannot be changed retrospectively as, I think, the emergency linked to the pandemic has been overcome in Japan.

Author Response

Thank you for your critical comments.

We added the descriptions about the limited generalizability due to using only one scenario in the limitation section as below(L243-244) .

"Second, only a single scenario was used for the simulation; thus, generalizability of our finding is uncertain."